# Isolation and Enumeration of CTC in Colorectal Cancer Patients: Introduction of a Novel Cell Imaging Approach and Comparison to Cellular and Molecular Detection Techniques

**DOI:** 10.3390/cancers12092643

**Published:** 2020-09-16

**Authors:** Alexander Hendricks, Burkhard Brandt, Reinhild Geisen, Katharina Dall, Christian Röder, Clemens Schafmayer, Thomas Becker, Sebastian Hinz, Susanne Sebens

**Affiliations:** 1Department of General, Visceral, Thoracic, Transplantation and Pediatric Surgery, University Hospital Schleswig-Holstein Campus Kiel, Arnold-Heller-Str. 3, Building C, 24105 Kiel, Germany; alexander.hendricks@med.uni-rostock.de (A.H.); kathi.dall@live.com (K.D.); clemens.schafmayer@med.uni-rostock.de (C.S.); thomas.becker@uksh.de (T.B.); Sebastian.hinz@med.uni-rostock.de (S.H.); 2Institute of Clinical Chemistry, University Hospital Schleswig-Holstein Campus Kiel, Arnold-Heller-Str. 3, Building U30, 24105 Kiel, Germany; Burkhard.Brandt@uksh.de; 3ORGA Labormanagement GmbH, Hospitalstraße 2, 48607 Ochtrup, Germany; reinhild.geisen@email.uni-kiel.de; 4Institute for Experimental Cancer Research, Kiel University and University Hospital Schleswig-Holstein Campus Kiel, Arnold-Heller-Str. 3, Building U30 Entrance 1, 24105 Kiel, Germany; c.roeder@email.uni-kiel.de

**Keywords:** liquid biopsy, circulating tumour cells, colorectal cancer, NYONE^®^, isolation by size of epithelial tumour cells, ScreenCell^®^

## Abstract

**Simple Summary:**

Despite significant progress in screening and treatment regimens, colorectal cancer (CRC) still is a major health burden lacking profound liquid biomarkers for identifying patients at risk. Circulating tumour cells (CTC) have the potential to non-invasively improve the diagnosis. We have already established a sensitive semi-quantitative RT-qPCR against CK20 for CTC quantification in CRC patients. For clinical translation, this study aims to validate our molecular detection method by terms of cytological approaches, and implement a novel semi-automated microscopic detection after immunofluorescence labelling of CTC. Additionally, we aim to compare our PCR-based approach to a marker-independent, but size-dependent, enrichment process. We have successfully applied the validation techniques and proved their feasibility. Enumeration by size yielded the highest numbers of CTC and demonstrated to be the most reliable strategy for CTC detection in CRC patients. Future studies with larger patient cohorts will have to investigate the clinical significance and prognostic value of this approach.

**Abstract:**

Circulating tumour cells (CTC) were proven to be prognostically relevant in cancer treatment, e.g., in colorectal cancer (CRC). This study validates a molecular detection technique through using a novel cell imaging approach for CTC detection and enumeration, in comparison to a size-based cellular and correlated the data to clinico-pathological characteristics. Overall, 57 CRC patients were recruited for this prospective study. Blood samples were analysed for CTCs by three methods: (1) Epithelial marker immunofluorescence staining combined with automated microscopy using the NYONE^®^ cell imager; (2) isolation by size using membrane filtration with the ScreenCell^®^ Cyto IS device and immunofluorescence staining; (3) detection by semi-quantitative Cytokeratin-20 RT-qPCR. Enumeration data were compared and correlated with clinic-pathological parameters. CTC were detected by either approach; however, with varying positivity rates: NYONE^®^ 36.4%, ScreenCell^®^ 100%, and PCR 80.5%. All methods revealed a positive correlation of CTC presence and higher tumour burden, which was most striking using the ScreenCell^®^ device. Generally, no intercorrelation of CTC presence emerged amongst the applied techniques. Overall, enumeration of CTC after isolation by size demonstrated to be the most reliable strategy for the detection of CTC in CRC patients. Ongoing studies will have to unravel the prognostic value of this finding, and validate this approach in a larger cohort.

## 1. Introduction

Colorectal cancer (CRC) is an extensive health burden that, according to estimates, will account for >1 million annual deaths worldwide by 2030 [1]. Despite considerable ongoing improvement and progress in treatment and screening [2], the average five-year survival rate still is below 70%, and nearly a quarter of the patients show distant metastases at the time of diagnosis with dismal five-year survival rates below 20% [3].

In recent years, the importance of individualised diagnostics and therapeutic options yielded major attention [4,5]. Biomarkers for either early detection of cancer or proof of minimal residual disease have been identified [6], e.g., enumeration of circulating tumour cells (CTC). The major challenges for CTC detection are the rarity of CTC in the peripheral blood, the technique for their enrichment and the discrimination of these cells from the diverse leukocyte populations [7]. For detecting these rare events, multiple methodologies are available. In general, two major approaches are distinguishable: Direct CTC detection by cytological staining and imaging or indirect CTC detection by molecular approaches, as PCR-based techniques.

In the latter context, various PCR target sequences have been used for CTC detection, however, in recent studies, we were able to establish a highly specific and sensitive RT-qPCR approach for detection of cytokeratin 20 (CK20) expression, which is widely found in mature enterocytes and also commonly in CRC cells [8,9,10]. In general, the detection of CTC in peripheral blood identified CRC patients with an unfavourable prognosis which could also be demonstrated by our CK20 RT-qPCR based detection approach [8,11,12].

A wide range of cellular techniques for CTC isolation and detection involves the enumeration of cells, based on the expression of certain markers on the cell surface as it is, for example, performed by the already approved semi-automated CellSearch^®^ platform that deploys the utilisation of anti-EpCAM and anti-pan-cytokeratin (pan-CK) antibodies [11,13,14,15]. Though, especially in the early stages of CRC, the detection rates of CTC by the CellSearch^®^ system are limited [16]. To potentially overcome this issue, the employment of a wider range of antibodies for the enrichment and/or biomarker detection could be beneficial. To compare our molecular PCR based approach with a cytological detection method and to overcome this critical issue, we developed and implemented a workflow that allows for immunofluorescence (IF) staining with a more extensive range of antibodies (anti-EGFR, anti-Her2, anti-EpCAM and anti-panCK) and a semi-automated cell enumeration by the cell imager NYONE^®^ (SYNENTEC, Elmshorn, Germany).

Moreover, an ongoing debate about the sensitivity of antibody-dependent enumeration methods of CTC is held due to the process of Epithelial-Mesenchymal-Transition (EMT), by which epithelial tumour cells lose their polarity and become enabled to disseminate [17]. During EMT, tumour cells might lose specific epithelial marker proteins, which are often used for the detection of CTC, e.g., EpCAM and cytokeratin. Thus, enumeration of CTC by means of antibody-mediated staining of those antigens might lead to an underestimation of the CTC count [18,19]. Consequently, antigen-independent enumeration approaches, such as isolation by size of CTC, have been developed. A strategy for this has been developed by ScreenCell^®^ (Sarcelles, France) with the ScreenCell^®^ Cyto IS device. Here, blood samples are transferred through a porous membrane and CTC that are significantly larger in diameter compared to leukocytes are retained by that membrane. The utility demonstrating the prognostic relevance of CTC has already been depicted in various reports [20,21,22].

In this prospective study, we applied and compared three techniques for the detection of CTC in CRC patients to validate our established CK20 RT-qPCR based detection method. Firstly, we applied a novel semi-automated microscopical approach for cytological CTC detection by a cell imager (NYONE^®^, SYNENTEC, Elmshorn, Germany). For this purpose, cells were enriched, and the CTC fraction was labelled by immunofluorescence staining with specific antibodies against well-established marker antigens of CRC cells—pan-CK, EpCAM, EGFR and Her2. Secondly, for the isolation by size of epithelial tumour cells, we implemented CTC detection with a well-established marker-independent size-exclusion method by the ScreenCell^®^ Cyto IS device which was combined with subsequent immunofluorescence staining of pan-CK and the leukocyte marker CD45. Finally, as a third method, the CK20 RT-qPCR was utilised for indirect CTC detection, which was already applied in previous studies on CRC patients [8,9,10]. Our primary aim was to validate our well-established PCR technique for CTC quantification by means of two cytological detection approaches. We further intend to analyse and elucidate the variances of two differing CTC enumeration methodologies by both size and marker dependent and independent concepts. Additionally, we intend to foster whether these approaches correlate with clinical parameters and with each other, to identify the most reliable CTC isolation and enumeration method.

## 2. Results

### 2.1. Patients and Demographics

In total, 57 patients with a histopathologically confirmed colorectal adenocarcinoma were included in this prospective study. The assessed cohort comprised 21 female and 36 male patients. Thirty-eight patients were diagnosed with colon carcinomas and 19 patients with rectal carcinomas. In the subset of colon cancer patients, the group was further stratified in right-sided colon cancer (17 patients) and left-sided colon cancer (21 patients). The median age of the entire study population at the time of blood withdrawal was 66 years (range: 42–89 years). In total, this study compared the detection of CTC in the blood of CRC patients by the use of three methods: (1) IF staining and semi-automated microscopical enumeration by the cell imager NYONE^®^; (2) isolation via the ScreenCell^®^ Cyto IS device coupled with cytochemistry according to Pappenheim and IF staining followed by microscopical enumeration; (3) semi-quantitative CK20 RT-qPCR.

Forty-four patients were enrolled for the semi-automated microscopical detection by the cell imager NYONE^®^. The median age of this NYONE^®^-cohort was 66 years (range: 45–89 years) with 32 patients diagnosed with colon cancer and 12 patients with a rectum carcinoma. In 31 cases blood samples were available for the analysis of CTC by the ScreenCell^®^ Cyto IS device. Within this subset, the majority of patients were male (21 male and 10 female patients) and diagnosed with colon carcinomas (22 colon vs 9 rectum carcinoma). The median age and range at the time of blood drawl were equal to the general cohort with 66 years (range: 42–89 years). Furthermore, the distribution, according to the tumour stages, was similar to the overall study population, with the majority of patients being diagnosed with stage three disease. Forty-one patients were enrolled for CTC detection by semi-quantitative CK20 RT-qPCR, previously reported [8,9,10]. The median age of this subset was 68 years (range: 45–89 years). Dissemination of patients across gender, tumour site and tumour stage were in general similar to the whole study population. Table 1 displays a full synopsis of all clinical and demographical data of the analysed patient cohort, as well as the employed methods for the detection of CTC.

### 2.2. Spiking Experiments and Validation of Cytological and RT-qPCR Detection Techniques

For validation of the employed detection techniques in this study, the human CRC cell line HT29 was utilised. Spiking experiments with HT29 cells in the blood of healthy donors were already successfully conducted for validating the detection of CTC in CRC patients by CK20 RT-qPCR [8]. To set up and validate the automated microscopic detection of CTC by the NYONE^®^ device, HT29 cells were also spiked into healthy donors’ blood. After isolation of the MNC-fraction using Vacutainer-CPT tubes, staining of the samples with anti-EGFR, anti-Her2, anti-EpCAM, anti-pan-CK antibodies conjugated with Alexa647 (red) and Alexa488-conjugated anti-CD45 antibodies (green), as well as staining of the nuclei using DAPI, a strong fluorescence signal for either CTC (red) or leucocytes (green) could be detected (Figure 1A), demonstrating that the CTCs were sufficiently distinguishable from leucocytes using the image processing YT^®^-Software (SYNENTEC, Elmshorn, Germany).

For validation of the ScreenCell^®^ Cyto IS technique, again, HT29 cancer cells were spiked into healthy donors´ blood samples. Similar to the NYONE^®^-approach, a strong green immunofluorescence signal of CTC after staining with the anti-pan-CK antibody could be observed, and leucocytes showed an exclusive strong red signal after staining with anti-CD45 antibodies. Additionally, leucocytes were significantly smaller in size compared to the HT29 cells providing another parameter for discrimination of CTC from PBMC (Figure 1B).

### 2.3. Detection of CTC by Automated Microscopy with the Cell Imager NYONE^®^

During the study, 44 patients were enrolled in this study arm (Table 1). In 16 patients (36.4%) ≥1 CTC were detected. Analysing the entire cohort, the mean amount of CTC was 0.89 cells (range: 0–7 cells; SD: 1.57). Figure 2A exemplarily depicts the CTC detection by NYONE^®^ in patients´ samples. Examining the study cohort and stratifying by demographical and clinical parameters, no significant difference in the quantity of CTC occurrence was detected in dependence on gender, age, tumour site or tumour localisation within colon carcinoma (Figure 3A and Table 2). However, a higher mean CTC count by trend was observed in patients with advanced tumour stages (UICC III + IV) compared to UICC stages I + II (1.06 cells vs 0.58 cells; *p* = 0.503) (Table 2).

### 2.4. Capture and Detection of CTC by ScreenCell^®^ Cyto IS Device

In all cases, the technical application of the ScreenCell^®^ filtration approach was successful. In 100% of the blood samples CTCs could be detected. All over the study population, the mean count for detected CTC was 3.25 CTC/mL (range: 0.2–14.3 CTC/mL; SD: 3.10) (Table 2). In Figure 2B, CTC from patients´ blood samples were enriched by the ScreenCell^®^ Cyto IS device and IF stained for detection. No statistical significance was found among the subsets of gender, age or tumour site (all *p* = ns) (Figure 3B and Table 2). Correlating the data with the relative tumour burden in compliance with the UICC stages, similar to the data obtained by the NYONE^®^ technique, patients with advanced disease (stage III and IV) exhibited significantly more CTC compared to patients with UICC stage I + II (mean: 4.10 CTC/mL vs 2.35 CTC/mL; *p* = 0.039) (Figure 3B).

### 2.5. Relative Quantification of the CTC Load by CK20 RT-qPCR

Finally, we determined the CTC load in our patient cohort with the well-established CK20 RT-qPCR [8,9,10]. In total, blood samples from 41 patients were collected (Table 1). In 33 cases (80.5%), the PCR revealed positive CK20 signals with a mean of relative CK20 mRNA expression units [EU] of 3.11 (range: 0–21.99 [EU]; SD: 3.81) (Table 2).

As shown in Figure 4C, similar results as revealed by the two methods of cytological enumeration of CTC were obtained. No significant differences were seen in the quantification of CTC by means of gender, age, tumour site, tumour localisation within the subset of colon cancer patients and the tumour stages (Figure 3C and Table 2). Interestingly, analysing the patients according to the tumour stages, an almost inverse relative detection of CK20 positive CTC was observed—with the mean relative CTC detection of stage I + II patients being 3.54 [EU] (range: 0–21.99 [EU]; SD: 4.70) vs 2.71 [EU] (range: 0–9.80 [EU]; SD: 2.76) in stage III + IV patients (*p* = 0.491) (Figure 3C). In the subset of stage I patients, one patient´s blood sample showed exceptionally high EU of CK20 mRNA (21.99 [EU]), potentially causing a significant bias to the analysis. Considering this as an outlier and re-analysing the data, the mean value of CTC detection in stage I + II patients was 2.57 [EU] (range: 0–5.88 [EU]; SD: 1.85) (data not shown). Thus, it can be concluded that the trend of lower relative CTC measurements in early stages compared to later stages could also be seen in this method (*p* = 0.854, data not shown).

### 2.6. The Coherence of Applied Detection Methods

To further compare and validate the applied CTC detection techniques, blood samples from 21 patients were analysed with all three techniques. In 11 and 15 patients, respectively, CTC detection by the NYONE^®^ cell imager and the CK20 RT-qPCR was possible, and in all cases, we were able to detect CTC applying the ScreenCell^®^ Cyto IS device.

First, the data regarding the correlation of the two cytological approaches are showing a quite heterogenous picture. There was no significant correlation of the overall CTC count between both methods. Thus, a high CTC count obtained by the ScreenCell^®^ Cyto IS device did not cohere with a high CTC count in the NYONE^®^ approach (r = 0.251; 95% CI: −0.192 to 0.609; *p* = 0.248) (Figure 4A). A more detailed analysis revealed that there is an inclination of the mean CTC count with advancing tumour stages in both the NYONE^®^ and the ScreenCell^®^ CTC enumeration techniques (mean: 0.42 cells in UICC I + II vs mean: 1.56 cells in UICC III + IV; *p* = 0.017 and mean: 2.08 cells in UICC I + II vs mean: 4.26 cells in UICC III + IV; *p* = 0.148, respectively), though only within the subset of CTC detected by the NYONE^®^ cell imager there is a statistical significance (Table 3). Data correlating the CTC count with tumour site characteristics differed for both cellular detection methods. In samples analysed with the NYONE^®^ cell imager, the mean CTC count for rectum carcinoma patients was 1.57 cells, and for colon carcinoma patients significantly less: 0.57 cells (*p* = 0.045). By applying the ScreenCell^®^ Cyto IS device for the same subset of patients, the mean count of CTC in rectum carcinoma patients was 2.03 cells, and in colon cancer patients with 3.51 cells higher by trend (*p* = 0.356). However, within colon cancer patients (right vs left), the data were congruent. There was a trend towards higher cell count in patients with left-sided colon carcinoma being detected by either detection method (Table 3).

Next, the techniques for cytological enumeration of CTC were compared with the molecular approach using CK20 RT-qPCR. Similar as described for the two cytological methods, no significant positive correlation between the enumeration results of either the NYONE^®^ or the ScreenCell^®^ technique with the CK20 RT-qPCR could be determined (r = 0.133; 95% CI: –0.170 to 0.448; *p* = 0.154) and (r = 0.339; 95% CI: −0.122 to 0.680; *p* = 0.337) (Figure 4B,C). Interestingly, analogous to the PCR data described above, the relative mean count for CTC (expressed by [EU]) in patients grouped by their tumour stages was even though not statistically significant (*p* = 0.521) opposing with the mean [EU] in stage I + II patients being 3.50 [EU] and in stage III + IV patients being 2.10 [EU] (Table 3). Again, regarding the stage I patient with an exceptionally high relative CTC count as an outlier, re-analysis of the data revealed a more consistent outcome by trend—stage I + II patients showed a mean value for relative CTC detection of 1.81 [EU] and stage III+IV patients 2.10 [EU] (*p* = 0.665, data not shown). Regarding the tumour site colon vs rectum, analogous to the cytological CTC detection technique with the ScreenCell^®^ Cyto IS device, there was a trend for a lower relative CTC count for patients suffering from rectal carcinoma (mean 2.33 [EU]) compared to patients with colon cancer (mean 3.18 [EU]) (*p* = 0.709). Analysing the colon cancer patients in more detail and stratifying for right-sided and left-sided colon cancer, interestingly and contrary to both cytological CTC detection methods, there was a trend towards more CTC being detected in right-sided colon cancers (right: mean 4.22 [EU] and left: mean 2.15 [EU]; *p* = 0.517).

## 3. Discussion

Despite perspicuous progress in the field of diagnosis and therapeutic efficiency in recent decades, CRC still raises various obscurities. It still is a major health burden and patients still often die, due to disease progression because informative biomarkers for monitoring the course and identifying early signs of progression are missing. Liquid biopsies have the potential to considerably revolutionise the scope of unique diagnosis and a precise follow-up by non-invasive means. Further, in recent years attention to biomarkers used for individualised diagnostics and therapeutic options has significantly increased. In this matter, various reports on the impact of CTC as a predictive and prognostic biomarker have been published so far [6,7]. CTC are thought to be directly linked to the primary tumour, detached from the cell bond, and hence, having the potential of initiating distant metastasis. After the process of intravasation, these CTC can be detected in peripheral blood by diverse approaches. However, CTC are extremely rare, and estimates are at about one cell per billion blood cells in patients with advanced disease stages [23].

In this prospective study, we deployed three discriminative techniques with the aim of associating the feasibility and plausibility of CTC determination related to clinico-pathological characteristics and to validate our already CK20 RT-qPCR based detection strategy. Thus, we established and later implemented a novel technique using the NYONE^®^ cell imager for the cytological enumeration and detection of CTC in CRC patients. This technique offers the potential of an easily reproducible and robust semi-automated microscopy-assisted cell count of CTC based on prior enrichment of PBMC and an IF staining with target-specific antibodies. A major benefit of this technique is the straightforward application process. Shortly after blood drawl, the samples are processed to prevent a significant loss of CTC, due to a potentially short CTC half-life [24], but hereafter the cells are fixed and can be stored for up to four days. This simplifies the operational sequences for the investigator significantly as it stores the patients´ samples and later simultaneous analysis of a larger sample cohort. For IF staining, we utilised the antibodies anti-pan-CK, anti-Her2, anti-EGFR and anti-EpCAM that were previously depicted to be specific for the detection of CTC of epithelial tumours and particularly CRC [25,26,27]. Many other studies also utilising IF staining for enumeration purposes, only apply one, or very few IF-coupled antibodies for staining [27,28,29]. This may though potentially cause a significant underestimation of the actual CTC count, due to non-detection of unstained CTC, leading to a bias in the samples. To evade this detriment and elevate the sensitivity of enumeration, we utilised a combination of IF-coupled antibodies. After IF staining, we utilised a semi-automated microscopical approach for the CTC enumeration. The cell imager´s software depicted possible positive events in terms of CTC, and the investigator was later presented these picture files for manual assessment. This significantly reduced the costs for personnel, and further theoretically limits error margins considerably, by waiving manual cell counting. Added values are by examination of cell morphology.

The method presented with a low CTC count (mean <1 cells, range 0–7 cells) and a moderate sensitivity of 36.4% (16/44) which might be not an improvement for enumeration and detection of CTC in peripheral blood samples of CRC patients. Nevertheless, in this study, we prospectively recruited a representative cohort of CRC patients across all stages of tumour progression, demonstrating higher CTC counts with increasing tumour stages. Despite lacking statistical significance, which might be reasonable for a number of CRC patients, a presumed interrelation between the tumour burden and the CTC count can be drawn. Most other CTC studies focus on patients with advanced disease—stage III + IV patients with suspected significantly higher detection rates of CTC. A previous study by Bork et al. has also analysed CTC in non-metastatic CRC patients by means of CellSearch^®^ technique and reported on exceptionally low rates of CTC in early tumour stages (≥2 CTC in 3.1% and ≥3 CTC in 1.7% of patients) and the lack of association of primary tumour characteristics with CTC detection [16]. Certainly, a downside, and hence, limitation of the marker dependent CTC detection as applied by Bork et al. with the widely employed CellSearch^®^ application is the potential underestimation of the total CTC count by omitting CTC that might not express EpCAM, due to preceding EMT [18,19]. We limited this drawback skipping the enrichment step by immunobeads and additionally applying EMT markers like EGFR [30]. Henceforth, a relevant subpopulation of CTC might be undetected and left out also by our technique.

To overcome this potential pitfall of underestimating CTC, our aim was to employ a marker- and antigen-independent physical enumeration technique to the same patient cohort as analysed by the NYONE^®^ cell imager approach. For this purpose, we reverted to the ScreenCell^®^ Cyto IS device. This is a technically simple to handle and cost-effective device for label-free isolation of CTC. The blood samples are passed through a membrane allowing for erythrocytes and leukocytes to pass through. Larger and less deformable cells, such as CTC, are effectively retained by the membrane, thus allowing for their enrichment and quantification. However, to clearly discriminate enriched tumor cells from leukocytes, we combined this size-dependent enrichment approach with a subsequent immunofluorescence labelling with anti-pan-CK and anti-CD45 antibodies. Our present data prove the positive surplus of CTC capture by the label-free isolation compared to the marker-dependent approach: The mean count for CTC in the samples analysed with the ScreenCell^®^ technique was more than threefold higher than in the NYONE^®^ subset. In the study conducted by Nicolazzo et al., the ScreenCell^®^ technique was compared to the CellSearch^®^ method as a label-dependent concept. Compliant to our findings, the marker-independent conception proved to be superior to the antigen-dependent technique of CTC enumeration, as significantly more CTC were captured by the ScreenCell^®^ method [31]. Moreover, we were able to positively correlate the clinical characteristics of the patient cohort to the CTC count, making our data more robust. With progressing tumour stages, the CTC load significantly increased, indicating that patients with a high tumour burden contain notably more CTC in peripheral blood, which in general is concordant with other studies of CRC patients [16]. The clinical value of high CTC numbers in our patients will be subsequently evaluated in another study, as soon as appropriate follow-up data are available.

Strategies for CTC enumeration relying on isolation by size, though potentially also do not harvest the CTC population entirely. There is a wide variability to the size of CTC [7], making smaller CTC more likely to be missed. Furthermore, in some cases, a significant contamination of leukocytes may negatively influence the ability of diligent CTC enumeration. Thus, our approach combining a size-dependent but marker-free CTC enrichment with subsequent immunofluorescence staining of CTC and leukocyte related antigens seems to enhance specificity and sensitivity of CTC detection and enumeration. Moreover, this approach yielded even a higher CTC detection rate as obtained by our well-established semi-quantitative CK20 RT-qPCR, which demonstrated already a clinical significance of indirect CTC detection in previous studies [8,9,10]. Importantly, the data concerning tumour burden and CTC load were concordant with the cytological approaches, as well as with a previous study that analysed CTC by CK20 RT-qPCR in a larger cohort of CRC patients [8].

## 4. Materials and Methods

### 4.1. Patient/Proband Recruitment and Sample Preparation

In total, 57 patients with a histologically verified colorectal carcinoma were prospectively enrolled in this study in the years 2017 and 2018. All patients underwent surgery at the Department of General, Visceral, Thoracic, Transplantation and Paediatric Surgery, University Hospital Schleswig-Holstein, Campus Kiel. Patients with UICC stage III or IV cancer were recommended to receive adjuvant or palliative chemotherapy, respectively, according to the therapy guidelines. All patients gave written informed consent to participate in this study, and the study was approved by the local ethics committee of the Medical Faculty, University of Kiel and the University Hospital Schleswig-Holstein, Campus Kiel (Reference No. A110/99). Classification of the pathological tumour stage was handled by the Department of Pathology, University Hospital Schleswig-Holstein, Campus Kiel, according to the TNM-classification (eighth edition). Clinical data were obtained from the clinical research database of the oncological biobank of the Comprehensive Cancer Center Kiel (BMB-CCC), and data were verified by re-examination of original patient records.

The peripheral blood samples were taken shortly prior to surgery from a central venous line. As three differing techniques for CTC detection were applied, the blood sample collection was handled optimally for the deployed method. For the semi-automated detection of CTC, blood was drawn into an 8.2 mL Citrate-Monovette (S-Monovette^®^ 8.2mL 9NC, 3.2% tri-Sodium Citrate, Sarstedt, Nümbrecht, Germany). For analysis with the ScreenCell^®^ Cyto IS device (ScreenCell^®^, Sarcelles, France) approximately 8 mL blood were drawn into an EDTA vacutainer (Vacutainer Tube EDTA (K2E), Becton Dickinson (BD), Heidelberg, Germany). For PCR analysis, approximately 20 mL blood were drawn with lithium heparin Monovettes (Sarstedt). All samples were further processed for analysis within 2 h.

### 4.2. Sample Analysis by IF Staining and Semi-Automated Microscopy—NYONE^®^

To validate the semi-automated microscopic approach with the NYONE^®^ (SYNENTEC, Elmshorn, Germany), cultured HT29 human CRC cells (approximately 100 cells, achieved by repeated counting) were spiked into 8.2 mL of blood from healthy donors who gave written informed consent. These blood samples were then transferred into Vacutainer-CPT-tubes (BD) and processed according to the manufacturer´s guidelines. The enriched mononuclear cell (MNC)-fraction was later incubated and stained with Alexa488-conjugated anti-CD45 antibodies (#304017; Biolegend, San Diego, CA, USA) for the detection of leucocytes (green fluorescence) and Alexa647-conjugated anti-EGFR (#sc-120 AF647; SantaCruz, Dallas, TX, USA), anti-Her2 (#3244412; Biolegend), anti-EpCAM (#324212; Biolegend) and anti-pan-CK (#628604; Biolegend) antibodies against the CTC (red fluorescence). After a washing step, a buffer containing DAPI (#422801; Biolegend, San Diego, CA, USA) was added, and automated microscopy was performed using the NYONE^®^ cell imager using the software package YT-software (SYNENTEC, Elmshorn, Germany) (Figure 5A). A CTC was defined as being DAPI and Alexa-647-positive, as well as Alexa488-negative. A detailed protocol of the method is given in the Appendix A.

### 4.3. Sample Analysis by Size-Dependent Filtration and IF Staining—ScreenCell^®^

For establishing the ScreenCell^®^ filtration device and testing specificity of the filtered tumour cells, HT29 cells were spiked into healthy donors´ blood (see above) and enriched on the isolation support (IS) with ScreenCell^®^ Cyto (Figure 5B). A description of the workflow in full detail is given in the Appendix A. Briefly, the filters were stained with RAL555 (May-Grunwald-Staining, MGG) (RAL Diagnostics, Martillac, France) and analysed by an independent cytopathologist. For verification of the putative cancer cells detected by MGG staining, IF staining and microscopy was performed afterwards. After destaining of the cells, double IF immunostaining with the primary mouse anti-pan-CK (AE1/AE3, #M3515; Agilent, Santa Clara, CA, USA) antibodies against CTC and rabbit anti-CD45 (EP68) antibodies (#AC-0065A, Epitomics, Abcam, Cambridge, GB) against leucocytes, was carried out. Lastly, the secondary antibodies goat-anti-mouse and goat-anti-rabbit conjugated with Alexa488 (green—against CTC) (#A11001; Life Technologies, Carlsbad, CA, USA) and Alexa568 (red—against leucocytes) (#A11011; Life Technologies), respectively, were added. Note that the IF colours scheme of CTC and leucocytes of this protocol were contrary to the staining protocol of the NYONE^®^ technique. A CTC was defined as being DAPI and Alexa-488 positive, as well as Alexa568-negative. A detailed protocol of the method is given in the Appendix A.

### 4.4. Sample Analysis by Molecular Analysis of mRNA—Semi-Quantitative CK20 RT-qPCR

The application of a semi-quantitative RT-qPCR against CK20 has been previously established in our group [8]. Briefly, patients´ blood samples were processed by centrifugation through a Ficoll-Hypaque density cushion (GE Healthcare/Merck, Darmstadt, Germany) according to the supplier’s recommendation for the enrichment of the mononuclear cell (MNC) fraction. MNC-RNA was isolated with RNAPure^TM^ reagent (VWR Peqlab, Darmstadt, Germany) and cDNA was obtained by reverse transcription of 3 μg total RNA (Maxima First Strand cDNA Synthesis Kit, Thermo Fisher Scientific, Darmstadt, Germany). The qPCR assays were run in total volumes of 20 μl on 96-well plates (Sarstedt) using TaqMan gene expression assays for cytokeratin 20/KRT20 (CK20), Hs00966063_m1 and for the housekeeping gene TBP (TATA-box binding protein), Hs00427620_m1, as a reference in combination with the TaqMan Fast Advanced Master Mix on a StepOne Plus real-time PCR System (all ThermoFisher Scientific, Waltham, MA, USA). All samples were run in triplicate. Relative gene expression was calculated as arbitrary expression units (EU) by a simplified ΔC_t_ method based on the difference between CK20- and the reference gene TBP-C_t_ values computed using the StepOne software (ThermoFisher Scientific, Waltham, MA, USA).

### 4.5. Statistical Analysis

All reported *p*-values are two-sided and were regarded as statistically significant at *p* < 0.05. When a Gaussian distribution of the data was assumed, the parametric data were analysed by a *t*-test. Non-parametric data were analysed by a Mann-Whitney U-test. For analysis of the correlation of the CTC detection results of the different detection methods, the Pearson correlation coefficient was calculated. Statistical calculation and testing were performed with GraphPad Prism 8 (GraphPad Software, San Diego, CA, USA).

## 5. Conclusions

The present study depicts and proves the feasibility of three different methods for CTC detection and enumeration in CRC patients across all tumour stages. By the introduction of the semi-automated microscopy approach with the NYONE^®^, we implemented an investigator independent microscopy procedure for CTC detection that applies a set of four markers possibly boosting the sensitivity of CTC detection compared to already existing methods. However, this approach resulted in the lowest detection rate, while isolation of CTC by size (as a label-free technique with subsequent immunofluorescence labelling) yielded the highest rates of detection which was slightly higher than those indirectly obtained by CK20 RT-qPCR. All methods revealed a definite trend to rising CTC counts with advancing tumour burden. Since the primary aim of this study was the implementation of two cytological CTC detection techniques for validation of our molecular detection approach, the sample size of this prospective study is limited. For a recommendation for clinical use, and to substantiate the clinical implication of these results, has to be further supported, proficient follow-up data of this prospective study has to be collected, and a study with a larger cohort is required.

## Figures and Tables

**Figure 1 cancers-12-02643-f001:**
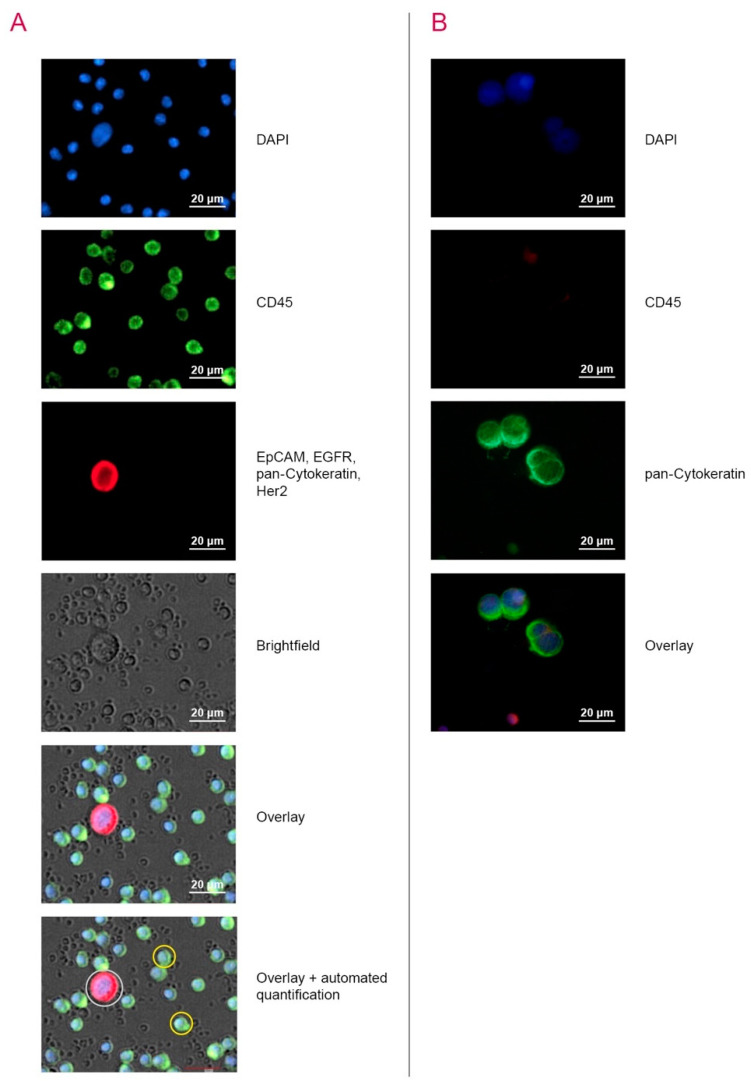
Validation of circulating tumour cells (CTC) detection by spiking experiments with HT29 colon cancer cells. HT29 cells were spiked into healthy donors´ blood samples. (**A**) NYONE^®^—After sample preparation, HT29 cells (red) were stained and identified with the NYONE^®^ cell imager and marked for automated quantification. Leukocytes were stained in green (yellow encirclement) and not considered for quantification. (**B**) ScreenCell^®^—HT-29 cells were immunofluorescence (IF) stained with anti-pan-CK antibodies (green) and leukocytes were IF stained with anti-CD45 antibodies (red). Detected HT29 cells were significantly larger compared to leucocytes.

**Figure 2 cancers-12-02643-f002:**
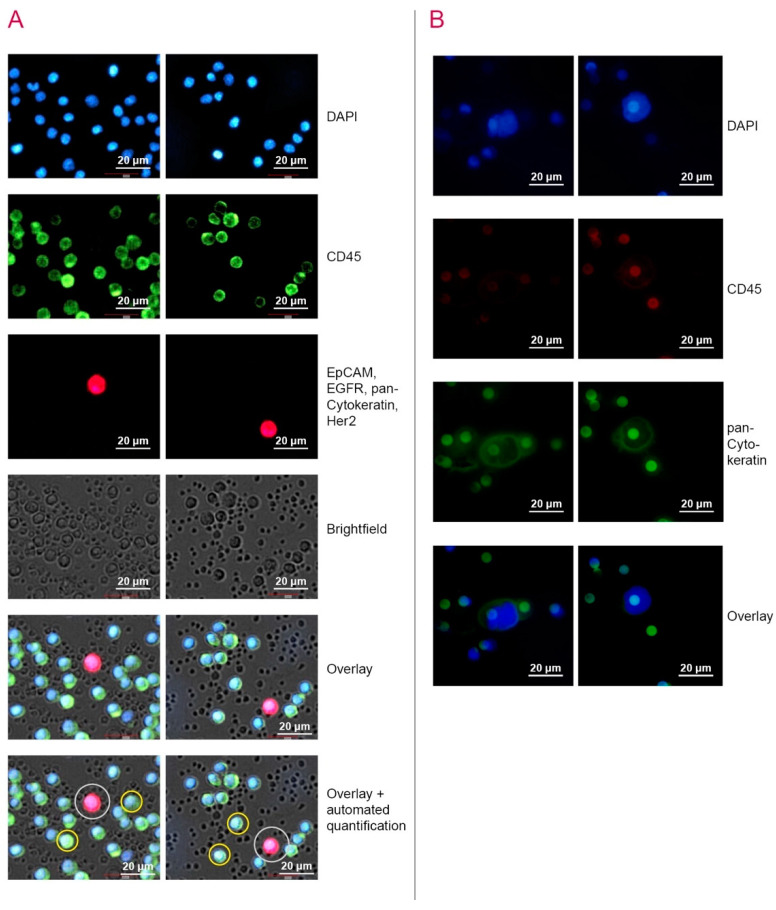
CTC detection in patient samples using NYONE^®^ and ScreenCell^®^ technology. CTC in peripheral patients´ blood samples were enriched by CPT vacutainer tubes for NYONE^®^ analysis, or by isolation by size via a porous membrane in the ScreenCell^®^ study cohort. Samples were then stained for CTC detection. (**A**) NYONE^®^—CTC were stained with anti-EpCAM, anti-EGFR, anti-Her2 and anti-pan-CK antibodies (red), and leukocytes were stained with anti-CD45 antibodies (green). DAPI staining was performed for nuclei staining (blue). Cells were scanned by NYONE^®^ and quantified by YT^®^-Software (here, CTC encircled in white and two exemplary CD45-positive cells in yellow) (**B**) ScreenCell^®^—CTC were IF stained with anti-pan-CK antibodies (green), leucocytes IF stained with anti-CD45 antibodies (red)and DAPI staining was performed for nuclei staining (blue).

**Figure 3 cancers-12-02643-f003:**
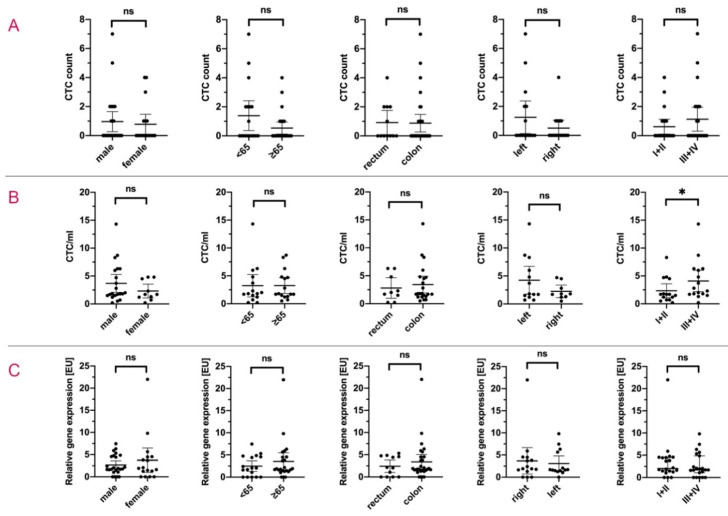
Congruence of CTC quantity and clinico-pathological characteristics. CTC enumeration data of the entire patient cohort was assessed by (**A**) NYONE^®^, (**B**) ScreenCell^®^ and (**C**) CK20-qRT-PCR and analysed in terms of the association between the prevalence of CTC and clinico-pathological data. The bar between the percentiles represents the mean value for CTC detection within each subset of analysed samples. <65 and ≥65 refers to the patients’ age in years at the time of blood drawl; left and right refers to the site of colon cancer. * *p* < 0.05.

**Figure 4 cancers-12-02643-f004:**
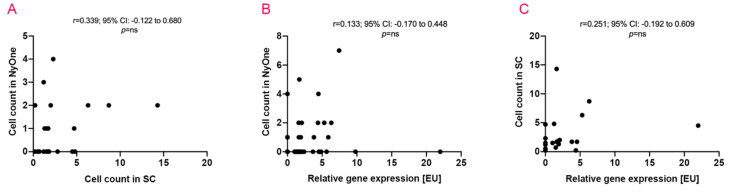
Correlation of the utilised detection methods and CTC quantity. Data obtained by each mode of detection for the CTC quantity were inter-correlated to the other technique. (**A**) NYONE^®^ vs ScreenCell^®^(SC). (**B**) NYONE^®^ vs CK20-RT-qPCR. **(C)** CK20 RT-qPCR vs ScreenCell^®^.

**Figure 5 cancers-12-02643-f005:**
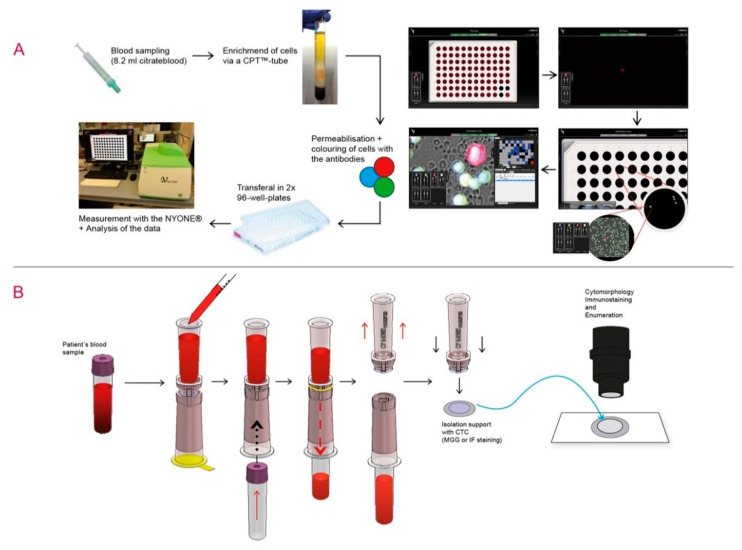
Experimental set-up for CTC detection using NYONE^®^ and ScreenCell^®^ technology. (**A**) NYONE^®^—Blood samples were collected. The enrichment of CTC was carried out by Ficoll centrifugation via CPT tubes. PBMC were fixed, permeabilised and stained with anti-CD45-Alexa488 (green to detect leukocytes), anti-EpCAM, anti-EGFR, anti-Her2 and anti-pan-CK antibodies (all Alexa647-coupled, red to detect epithelial cells) and DAPI (blue) for nuclei staining. The semi-automated enumeration was carried out by the cell imager NYONE^®^. Pre-scanning of all wells was done for the detection of red fluorescence, and all positive events were marked for further scanning for the detection of blue, green and red fluorescence, as well as a brightfield image. After image analysis, CTC (DAPI positive, negative for Alexa488 and positive for Alexa647) were encircled allowing cytological assessment. (**B**) ScreenCell^®^—Blood samples were collected with EDTA tubes. After adding the buffer solution, the sample was added to the filtration device. Adding an empty vacutainer, blood was drawn through the filter, and CTC remained on the filter. Followed by staining with MGG and/or IF, detection and enumeration of CTC were possible.

**Table 1 cancers-12-02643-t001:** Patient demographics and clinical characteristics of the entire study population and further breakdown according to the utilised detection modes.

Parameters	Total N (%)	NYONE^®^ N (%)	ScreenCell^®^ N (%)	CK20 RT-qPCR N (%)
	57 (100)	44 (100)	31 (100)	41 (100)
**Gender**				
Male	36 (63.2)	26 (59.1)	21 (67.7)	24 (58.5)
Female	21 (36.8)	18 (40.9)	10 (32.3)	17 (41.5)
**Age**				
Median (range)	66 (42–89)	66 (45–89)	66 (42–89)	68 (45–89)
<65	27 (47.4)	18 (40.9)	15 (48.4)	17 (41.5)
≥65	30 (52.6)	26 (59.1)	16 (51.6)	24 (58.5)
**Tumour site**				
colon	38 (66.7)	32 (72.7)	22 (71.0)	29 (70.7)
right	17 (44.7)	16 (50.0)	9 (40.9)	15 (51.7)
left	21 (55.3)	16 (50.0)	13 (59.1)	14 (48.3)
Rectum	19 (33.3)	12 (27.3)	9 (29.0)	12 (29.3)
**UICC stage**				
I	15 (26.3)	12 (27.3)	9 (29.0)	12 (29.3)
II	10 (17.5)	9 (20.5)	6 (19.4)	8 (19.5)
III	24 (42.1)	18 (40.9)	12 (38.7)	16 (39.0)
IV	8 (14.0)	5 (11.4)	4 (12.9)	5 (12.2)

Abbreviations: UICC—Union internationale contre le cancer.

**Table 2 cancers-12-02643-t002:** Association of clinico-pathological patients´ characteristics of the entire study population and CTC quantity partitioned for each technique of CTC detection.

Parameters	NYONE^®^			ScreenCell^®^			CK20 RT-qPCR		
	Positive *N* (%)	Mean (SD)	*p*	Positive *N* (%)	Mean (SD)	*p*	Positive *N* (%)	Mean (SD)	*p*
**Total**	16/44 (36.4)	0.89 (1.57)		31/31 (100)	3.25 (3.10)		33/41 (80.5)	3.11 (3.81)	
**Gender**									
Male	10/26 (38.5)	0.96 (1.71)	0.708	21/21 (100)	3.69 (3.52)	0.257	20/24 (83.3)	2.67 (2.19)	0.383
Female	6/18 (33.3)	0.78 (1.40)		10/10 (100)	2.32 (1.74)		13/17 (76.5)	3.74 (5.35)	
**Age**									
<65	8/18 (44.4)	1.39 (2.06)	0.078	15/15 (100)	3.23 (3.62)	0.980	12/17 (70.6)	2.48 (2.31)	0.379
≥65	8/26 (30.8)	0.54 (1.03)		16/16 (100)	3.26 (2.65)		21/24 (87.5)	3.56 (4.58)	
**Tumour site**									
Rectum	5/12 (41.7)	0.92 (1.31)	0.939	9/9 (100)	2.81 (2.39)	0.624	8/12 (66.7)	2.43 (2.20)	0.466
colon	11/32 (34.4)	0.88 (1.68)		22/22 (100)	3.43 (3.38)		25/29 (86.2)	3.40 (4.30)	
right	5/16 (31.3)	0.50 (1.03)	0.212	9/9 (100)	2.26 (1.46)	0.182	12/15 (80.0)	3.69 (5.37)	0.713
left	6/16 (37.5)	1.25 (2.11)		13/13 (100)	4.24 (4.10)		13/14 (92.9)	3.08 (2.94)	
**UICC stage**									
I + II	3/12 (25.0)	0.58 (1.24)	0.503	15/15 (100)	2.35 (2.24)	0.039	11/12 (91.7)	3.54 (4.70)	0.491
III + IV	6/18 (33.3)	1.06 (1.89)		16/16 (100)	4.09 (3.60)		13/16 (81.3)	2.71 (2.76)	

Statistical analysis was performed using an unpaired *t*-test for parametric and a Mann-Whitney U-test for non-parametric data. All *p*-values in bold are regarded as statistically significant. Abbreviations: UICC—Union internationale contre le cancer; SD—standard deviation.

**Table 3 cancers-12-02643-t003:** Association of Clinico-Pathological Patients´ Characteristics and CTC Quantity of the Patients´ Subset Where all Three Tests Were Positive for CTC Detection. The Data is Partitioned for Each Technique of CTC Detection.

Parameters	NYONE^®^			ScreenCell^®^			CK20 RT-qPCR		
	Positive *N* (%)	Mean (SD)	*p*	Positive *N* (%)	Mean (SD)	*p*	Positive *N* (%)	Mean (SD)	*p*
**Total**	11/21 (52.4)	0.90 (1.03)		21/21 (100)	3.01 (3.16)		15/21 (71.4)	2.90 (3.81)	
**Gender**									
Male	7/14 (50.0)	0.86 (0.95)	0.785	14/14 (100)	3.34 (3.97)	0.541	10/14 (71.4)	2.16 (2.13)	0.328
Female	4/7 (57.1)	1.00 (1.41)		7/7 (100)	2.36 (1.69)		5/7 (71.4)	4.37 (7.88)	
**Age**									
<65	6/9 (66.7)	1.44 (1.33)	**0.047**	9/9 (100)	3.30 (4.50)	0.746	5/9 (55.6)	1.99 (2.22)	0.464
≥65	5/12 (41.7)	0.50 (0.67)		12/12 (100)	2.80 (2.40)		10/12 (83.3)	3.58 (6.04)	
**Tumour site**									
Rectum	5/7 (71.4)	1.57 (1.40)	**0.045**	7/7 (100)	2.03 (2.06)	0.356	4/7 (57.1)	2.33 (2.39)	0.709
colon	6/14 (42.9)	0.57 (0.76)		14/14 (100)	3.51 (3.84)		11/14 (78.6)	3.18 (5.66)	
right	3/7 (42.9)	0.43 (0.54)	0.502	7/7 (100)	2.24 (1.67)	0.834	4/7 (57.1)	4.22 (7.96)	0.517
left	3/7 (42.9)	0.71 (0.95)		7/7 (100)	2.03 (2.06)		7/7 (100)	2.15 (1.85)	
**UICC stage**									
I + II	4/12 (33.3)	0.42 (0.67)	**0.017**	8/8 (100)	2.08 (1.64)	0.148	7/8 (87.5)	3.50 (6.05)	0.521
III + IV	7/9 (77.8)	1.56 (1.24)		6/6 (100)	4.26 (4.65)		4/6 (66.7)	2.10 (2.28)	

Statistical analysis was performed using an unpaired *t*-test for parametric and a Mann-Whitney U-test for non-parametric data. All *p* values in bold are regarded as statistically significant. Abbreviations: UICC—Union internationale contre le cancer; SD—standard deviation.

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
