# Peer review of "Isolation and Enumeration of CTC in Colorectal Cancer Patients: Introduction of a Novel Cell Imaging Approach and Comparison to Cellular and Molecular Detection Techniques"

_cancers, 2020, doi:10.3390/cancers12092643_

Round 1
Reviewer 1 Report
Hendricks and colleagues conducted a comparative study aiming at demonstrating the validity of a novel cell imaging technique to detect CTC in CRC patients. I consider this as an exploration and add to current knowledge of this rapidly advancing field. I expect the comments below to be addressed. Major comments: a. I noticed that this study does not include a "Gold standard" as benchmark for other approaches. This makes it even harder to interpret and compare findings from these three methods. Could the authors explain why they did not adopt the CellSearch, which is by far the only approved approach, in the comparison? b.My primary concern lies in the blur of conclusion which is not clear enough in either the abstract or the end of the discussion. What is the recommendation for the novel technique? c. The major limitation of this study is the sample size, rendering the study underpowered to estimate the consistency across the three approaches. This ought to be highlighted in the text. d. The authors concluded that "Overall, enumeration of CTC after isolation by size demonstrated to be the most reliable strategy for the detection of CTC in CRC patients." This seems too optimistic to me. Is this possibly due to false-positive findings? One of the major limitations for this method is low specificity [PMID: 10623654], given that larger leukocytes may also be retained by the membrane. e. Abstract: I would suggest adding more detailed numbers, e.g. positive rates for different methods as this is important information for readers. Consider adding one sentence stating the strength of this novel technique. Minor comments: Table 1, Line 1, the entries of 100% in brackets are not necessary. Table 2. Consider adding a footnote clarifying what these p values stand for? What tests were used? The same applies to table 3. Although this might have been covered in the text, each table should be a stand-alone unit with full information. Figure 4, I would suggest adding correlation coefficients along with CIs in the figure.Author Response
Reviewer 1:
“Hendricks and colleagues conducted a comparative study aiming at demonstrating the validity of a novel cell imaging technique to detect CTC in CRC patients. I consider this as an exploration and add to current knowledge of this rapidly advancing field. I expect the comments below to be addressed. Major comments:”
Request 1: “I noticed that this study does not include a "Gold standard" as benchmark for other approaches. This makes it even harder to interpret and compare findings from these three methods. Could the authors explain why they did not adopt the CellSearch, which is by far the only approved approach, in the comparison?”
Answer 1: We fully apprehend and agree with the reviewer´s concern, however, important to note the aim of this study was not to compare our novel approach to the benchmark technique (CellSearch®). Since our group has successfully established the CK20 RT-qPCR for detection of CTC in CRC patients, this study aimed at validating this molecular detection method by terms of cytological approaches and implementing a novel semi-automated microscopic detection after immunofluorescence labelling of CTC. Additionally, we aimed to compare our PCR based approach to a marker-independent but size-dependent enrichment process.
To make this point clearer to the readership, we have modified the following sections of the manuscript: Abstract, page line 53/54; introduction (+ slightly restructured) Page 3, line 99/100 and line 115-129; discussion, page 11 line 302; new conclusion section, page 15, line 467-480.
Request 2: “My primary concern lies in the blur of conclusion which is not clear enough in either the abstract or the end of the discussion. What is the recommendation for the novel technique?”
Answer 2: We agree with the reviewer and have amended the conclusion paragraph accordingly: Page 15, line 461-474. We clearly stated our aim of the study and the relevance of our findings in this subsection.
Request 3: “The major limitation of this study is the sample size, rendering the study underpowered to estimate the consistency across the three approaches. This ought to be highlighted in the text.”
Answer 3: By addressing the reviewer`s former comment about the conclusion of our study, we also mentioned the limited sample size of our study in the conclusion section: Page 15, line 476-480.
Request 4: “The authors concluded that "Overall, enumeration of CTC after isolation by size demonstrated to be the most reliable strategy for the detection of CTC in CRC patients." This seems too optimistic to me. Is this possibly due to false-positive findings? One of the major limitations for this method is low specificity [PMID: 10623654], given that larger leukocytes may also be retained by the membrane.”
Answer 4: “We outright agree with the reviewer´s concern about the label-free approach with the ScreenCell® cyto IS. However, being aware of this issue, we combined this size-dependent enrichment approach with subsequent immunofluorescence stainings for CTC verification and validation (see also materials and methods, page 14, line 427-428). The applied antibody (pan-cytokeratin) for IF-staining is generally acknowledged for CTC detection of epithelial malignancies as CRC. Leukocytes were stained for CD45 so that CTC could be clearly discriminated and were classified as pan-CK+ and CD45-. To make this point clear to the readership we outline the experimental procedure in the introduction, page 3, line 122/123 as well as in our discussion, page 12, line 348 to page 13, line 369.
Request 5: “Abstract: I would suggest adding more detailed numbers, e.g. positive rates for different methods as this is important information for readers. Consider adding one sentence stating the strength of this novel technique.”
Answer 5: As requested, we added the overall positivity rates (%) of each applied technique to the abstract.
Minor comments:
Request 6: “Table 1, Line 1, the entries of 100% in brackets are not necessary.”
Answer 6: After discussing this point raised by the reviewer amongst the co-authors, for the consistency and format of the tables we consider the frequencies (even being 100%) to be necessary and valuable to the readership.
Request 7: “Table 2. Consider adding a footnote clarifying what these p values stand for? What tests were used? The same applies to table 3. Although this might have been covered in the text, each table should be a stand-alone unit with full information.”
Answer 7: We thank the reviewer for pointing this out. We added footnotes to Tables 2 and 3, stating which statistical tests were used, in order to amend the tables as stand-alone units.
Request 7: “Figure 4, I would suggest adding correlation coefficients along with CIs in the figure.”
Answer 7: As requested, we have modified the figures by mentioning the statistically derived values.
Reviewer 2 Report
In the manuscript titled, “Isolation and enumeration of CTC in colorectal cancer patients: Introduction of a novel cell imaging approach and comparison to cellular and molecular detection techniques”, the authors presented three different techniques for analysis of circulating tumor cell (CTC), and reported that CTC isolated based on the size provided the most reliable strategy.The manuscript would be interesting to the target audience and would be an important addition to the journal. The authors are recommended to check the on the flow of the manuscript.
Author Response
Reviewer 2:
“In the manuscript titled, “Isolation and enumeration of CTC in colorectal cancer patients: Introduction of a novel cell imaging approach and comparison to cellular and molecular detection techniques”, the authors presented three different techniques for analysis of circulating tumor cell (CTC), and reported that CTC isolated based on the size provided the most reliable strategy. The manuscript would be interesting to the target audience and would be an important addition to the journal.”
Request 1: “The authors are recommended to check the on the flow of the manuscript.”
Answer 1: We thank the reviewer for the overall positive statement and have carefully checked the manuscript.
Round 2
Reviewer 1 Report
My comments have been well addressed.
Reviewer 2 Report
In the updated version of the manuscript titled, "Isolation and enumeration of CTC in colorectal cancer patients: Introduction of a novel cell imaging approach and comparison to cellular and molecular detection techniques" the authors have taken in accounts the reviewer's comments and incorporated the recommended corrections.